# Collective action or individual choice: Spontaneity and individuality contribute to decision-making in *Drosophila*

Isabelle Steymans[1], Luciana M. Pujol-Lereis[2], Björn Brembs[1]*, E. Axel Gorostiza[1,3¤]*

**1** Institut für Zoologie - Neurogenetik, Universität Regensburg, Regensburg, Germany, **2** Laboratory of Amyloidosis and Neurodegeneration, Fundación Instituto Leloir, IIBBA, CONICET, Buenos Aires, Argentina, **3** Instituto de Fisiología, Biología Molecular y Neurociencias (IFIBYNE) CONICET - Universidad de Buenos Aires, Buenos Aires, Argentina

¤ Current address: Institute of Zoology, Biocenter Cologne, University of Cologne, Cologne, Germany
* eagorostiza@gmail.com (EAG); bjoern@brembs.net (BB)

**Data Availability Statement:** https://doi.org/10.6084/m9.figshare.13472259.v3.

## Abstract

Our own unique character traits make our behavior consistent and define our individuality. Yet, this consistency does not entail that we behave repetitively like machines. Like humans, animals also combine personality traits with spontaneity to produce adaptive behavior: consistent, but not fully predictable. Here, we study an iconically rigid behavioral trait, insect phototaxis, that nevertheless also contains both components of individuality and spontaneity. In a light/dark T-maze, approximately 70% of a group of *Drosophila* fruit flies choose the bright arm of the T-Maze, while the remaining 30% walk into the dark. Taking the photopositive and the photonegative subgroups and re-testing them reveals the spontaneous component: a similar 70–30 distribution emerges in each of the two subgroups. Increasing the number of choices to ten choices, reveals the individuality component: flies with an extremely negative series of first choices were more likely to show photonegative behavior in subsequent choices and *vice versa*. General behavioral traits, independent of light/dark preference, contributed to the development of this individuality. The interaction of individuality and spontaneity together explains why group averages, even for such seemingly stereotypical behaviors, are poor predictors of individual choices.

## Introduction

The hallmark of successful neuroscience is understanding nervous systems well enough to predict behavior [1–7]. Behavioral variability, the observation that even under extremely well-controlled experimental circumstances, animals will not always behave identically, has been the bane of this endeavor. The sources of this variability are numerous, from neural noise or sensory ambiguity to spontaneous variations in behaviors to either 'try out' behavioral solutions to a problem or to become unpredictable in competitive situations [8–11]. So far, largely because of this variability, neuroscience has mainly made progress predicting behavior on the population level, by averaging out some of this variability. However, even on the population

**Funding:** EAG and LMPL are members of the Argentinean National Scientific and Technical Research Council (CONICET). Experiments were financially supported by the University of Regensburg. The writing and review processes of this work were financially supported by CONICET, and an IBRO Return Home Fellowship. EAG current position at the University of Cologne is supported by NeuroNex (DFG grant Bu857-15/1).

**Competing interests:** The authors have declared that no competing interests exist.

level, there remain two sources of behavioral variability: inter-individual variability and intra-individual variability. Inter-individual variability may arise either from consistent differences between individuals (i.e., 'individuality' [12–23]), and/or by individuals generally behaving inconsistently over time (i.e., 'spontaneity'; [24–30]). Of course, individuals may also differ consistently in their variability [31]. This instantiation of the interplay between chance (spontaneity altering behavior from moment to moment) and necessity (character traits compelling individuals to behave consistently over time) is not well understood.

The attempt to reduce variability by averaging over multiple individuals in a group can potentially influence the behavior of the individuals in the tested population. For example, when tested alone, *rovers* (one of two variants of the *Drosophila foraging* gene) are more likely than *sitters* (the other variant) to leave a cold refuge in a heat maze apparatus. In contrast, this behavior was indistinguishable between *rovers* and *sitters* when tested in the same experiment, but in groups [32]. Conversely, individual *Drosophila* flies avoid an aversive odor only weakly, but in a group, this avoidance is strongly enhanced [33]. In another example, isolated *Periplaneta* cockroaches stay longer and more frequently in vanillin-scented than in unscented shelters, while groups choose more frequently the unscented shelters, because the smell of vanillin decreases the attraction between individuals [34]. Also important, group averaging may misrepresent individual behavior and could lead to false interpretation of the actual process. In honeybees, the reward-based olfactory conditioning of the proboscis extension response (PER) is commonly used to study classical and differential conditioning, as well as extinction. Here, the conditioned response probability at the population level features a gradual change during training, usually referred to as the learning curve. However, individual behavior is characterized by abrupt and stable changes in response probabilities, once the animal responded for the first time, it continues responding with a high probability in the subsequent trials, meaning that the group average learning curve due not represent a rise in the learning performance but rather an increase in the proportion of responding animals [35,36]. This type of artifacts in population average performances in learning paradigms has also been reported for vertebrates [37].

The preference shown by an animal for light (photopreference) is usually studied in the context of a special case, phototaxis, in which the animal moves towards or away from a source of light in the same direction of the source. Phototaxis is one of the most iconic insect behaviors typically described as rigid and stereotyped. Perhaps surprisingly, for this behavior, analogous differences between individual behavior and behavior in a group may exist. For instance, the individuality observed in *Drosophila* phototaxis [38] stands in apparent contradiction to an observation Seymour Benzer described as ". . . if you put flies at one end of a tube and a light at the other end, the flies will run to the light. But I noticed that not every fly will run every time. If you separate the ones that ran or did not run and test them again, you find, again, the same percentage will run. But an individual fly will make its own decision" (cited by [39]). Similar observations as in the quote were described for flies conditioned to prefer a particular odor [40] and recently mentioned again in [41]. In both cases, the setup used was similar to the classic countercurrent phototaxis paradigm (CPP) developed by Benzer, where flies have several opportunities in one session to approach a light source and the population ends up being split according to how many times each fly chose approaching the light [42]. Taken together, this literature suggests that flies in groups do not exhibit individuality, while they do when tested alone. Importantly, the literature negating individuality did not provide the underlying data, making it difficult to estimate the reliability of these statements. In order to tackle both the apparent contradiction and the lack of data, we tested *Drosophila* flies in groups for their photopreference in a light/dark T-maze, where flies can choose between getting into the illuminated arm of the T-Maze or the opaque one [43] and re-tested the photopositive and

photonegative subgroups separately, as it was described in the literature. In order to increase the number of decisions towards the high number of individual decisions described in [38], we also tested phototaxis proper in the classic CPP developed by Benzer [42]. As our decision-making tasks involved locomotion, we controlled for confounding effects of general activity parameters by testing individual flies from subgroups generated after a session of CPP experiments in Buridan's Paradigm, where flies walk between two opposing black stripes [44,45].

## Methods

### Strains and fly rearing

Flies were reared and maintained at 25 ˚C in vials containing standard cornmeal agar medium [46] under 12 h light/dark cycles with 60% humidity. All experiments were done with a Wild-type Berlin strain from our stock in Regensburg (*WTB*). For all experimental paradigms, 2-5d old flies were randomly chosen regardless of their sex. Pilot experiments support the notion that in the used paradigms, males and females behave indistinguishable from each other. For handling the animals, flies were always cold anesthetized for the minimal amount of time possible.

### T-Maze

We used the setup and conditions previously described in [43] (Fig 1A). Light/dark choice, i.e. photopreference, was measured in a custom-built, opaque PVC T-Maze with only one transparent (acrylic) choice tube (protocol DOI: 10.17504/protocols.io.c8azsd). Flies were placed in an initial dark tube (10 cm long, 1.5 cm inner diameter, and 2.5 cm outer diameter) and were left to dark adapt for 10 min. Then, they were transferred to the cylindrical elevator chamber (1.5 cm diameter, 1.5 cm height) by gently tapping the apparatus, where they remained for 30 s. Next, the elevator was placed between the dark and the bright tube (both 20 cm long, 1.5 cm inner diameter, and 2.5 cm outer diameter), and flies were allowed to choose for 30 s (Fig 1A). This time was adjusted according to the length of the choice tubes and the average speed of the flies according to previous experiments performed in the laboratory. The time is enough for a fly to reach the end of one tube, turn back, and cross to the other tube (i.e. changing the photopreference). However, longer choice periods do not affect the choice, as in our hands, flies showed similar preferences in this paradigm with 30 s or 3 min choice periods [43]. The light source was always placed 31.5 cm above the base of the T-Maze and consisted of a fluorescent warm white tube (OSRAM 18W/827), which delivers 1340 lux at that distance.

The Choice Index was calculated using the formula:

$$CI = \frac{\#F_L - (0.964 \times (\#F_D + \#F_E))}{\#F_T}$$

Where $F_L$ denotes the number of flies in the transparent tube, $\#F_D$ the number of flies in the opaque tube, $\#F_E$ the number of flies that were caught in the elevator, and $\#F_T$ the total number of flies. During pilot experiments, we observed that after a first choice session, there were insufficient flies in the dark tube and the elevator to separately re-test them in a second session, if the experiments required it. Therefore, for those experiments we pooled flies in two different ways in order to reach at least 50% of the flies tested in the first session (i.e, 40 flies). First, due to the fact that the elevator is also dark, we added the flies from the elevator to the ones in the dark tube. Second, we added flies from the dark tubes of independent replicates of first sessions performed on the same or consecutive days to reach at least 40 flies for a second choice session.

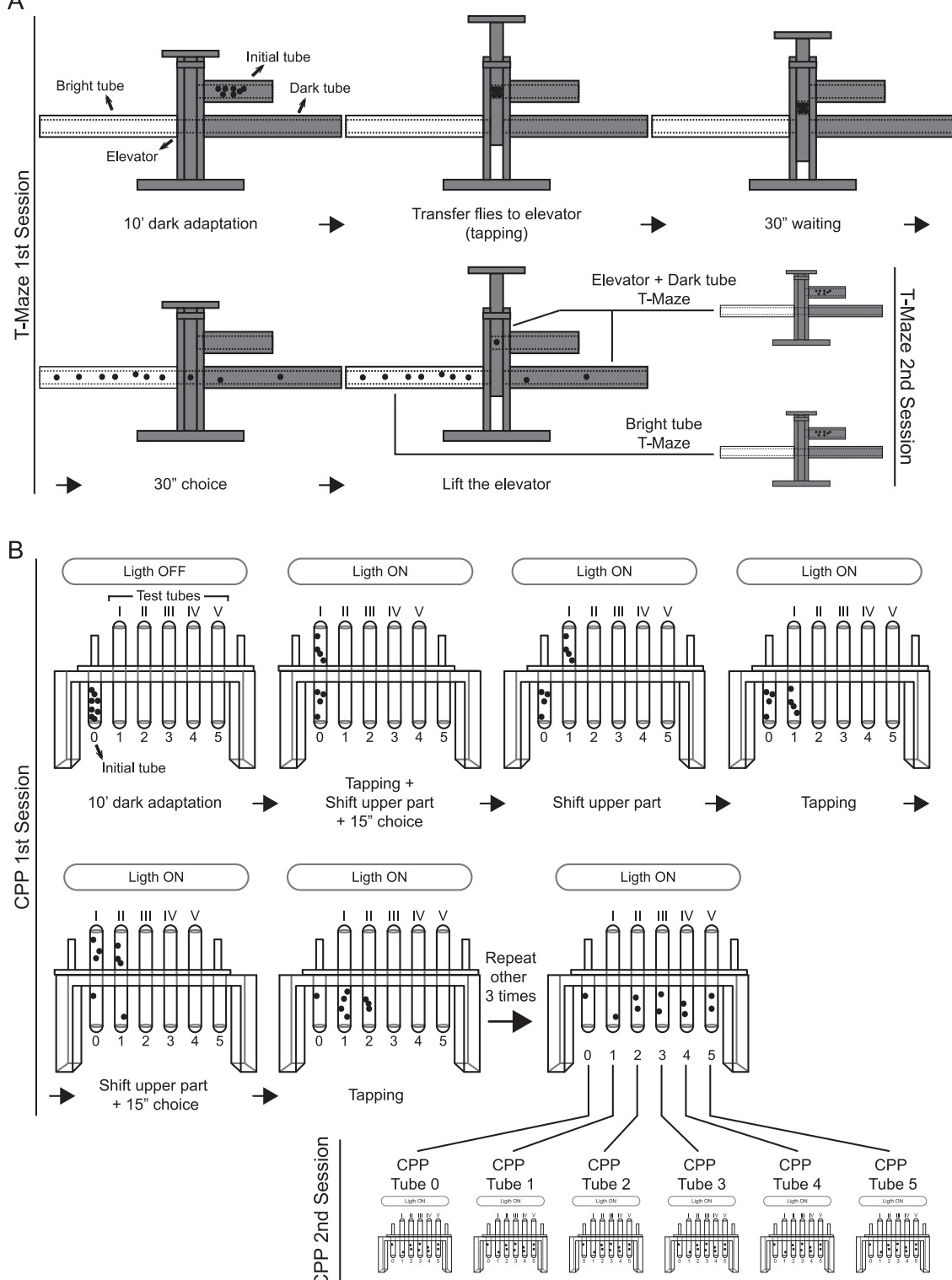

**Fig 1. Schematics of paradigms and experimental design.** Depiction of the first and second session of T-Maze experiments (**A**) and CPP experiments (**B**). The drawings show a single replicate for each first session. Sometimes, flies from different replicates were added to reach the minimum amount of flies required for a second session. This was especially the case for Elevator+Dark Tube in the T-Maze (**A**), and Tubes 0, 1 and 2 in the CPP (**B**).

Then, when the experiment required it, a second choice session in the T-Maze was separately performed by flies that chose the bright tube in the first session, and by a combination of flies from different replicates that chose the dark tube in the first session (Fig 1A). To compensate for the now slightly larger fraction of the T-maze being represented by the pooled tube and elevator, we corrected the CI by multiplying the number of flies that chose the dark and the elevator by 1–0.036 = 0.964, where 0.036 is the added proportion of darkness that the elevator represents. With this correction, a *CI* of 1 meant all the flies chose the bright arm, while an index of -0.964 meant all flies preferred the dark tube. The tubes were cleaned thoroughly after each session.

## Countercurrent phototaxis paradigm (CPP)

Phototaxis was evaluated using the CPP [42] (protocol DOI: 10.17504/protocols.io.c8gztv). In this paradigm, flies are separated according to five consecutive choices in a single session (Fig 1B). Each choice in this experiment involves either staying put or walking towards the light (positive phototaxis). The consequence of one such session with 5 consecutive choices is that the original group is split into six subgroups according to their sequence of choices: the flies choosing five times to walk towards the light end up in tube 5, while the ones never walking remain in tube 0. The apparatus is completely transparent and consists of two acrylic parts, a lower one with 6 parallel tubes (an initial tube named 0 + 5), and a movable upper part with 5 parallel test tubes. Each plastic tube has a length of 6.8 cm, an inner diameter of 1.5 cm, and an outer diameter of 1.7 cm. The test group was placed in the initial tube 0 and was left in darkness to acclimate for 10 min, with the apparatus placed horizontally. Thereafter, flies were startled by tapping the apparatus, making all of them end up at the bottom of tube 0. The apparatus was placed horizontally with the test tubes facing the light, and the upper part shifted, making tube 0 face the first test tube (tube I in Fig 1B) for 15 s, allowing the flies to move towards the light. This time was adjusted according to the length of the choice tubes and the average speed of the flies according to previous experiments performed in the laboratory. The time is enough for a fly to reach the end of one tube, turn back, and cross to the other tube (i.e. changing the photopreference). Then, the upper part was shifted again and flies that moved to the test tube were transferred to the next tube of the lower part by tapping the apparatus, and this cycle was repeated 4 more times. The light source was always placed at 30 cm from the apparatus and consisted of a fluorescent warm white tube (OSRAM 18W/827), which delivers 1340 lux at that distance.

The Performance Index was calculated using the formula:

$$\mathrm{PI} = \frac{(\#F_5 \times 5) + (\#F_4 \times 4) + (\#F_3 \times 3) + (\#F_2 \times 2) + (\#F_1 \times 1) + (\#F_0 \times 0)}{\#F_T}$$

Where $F_n$ was the number of flies in the tube n (being 0 the initial tube and 5 the last test tube), and $\#F_T$ was the total number of flies. After a first session of five such choices, there were usually less than 40 flies in each tube. In the tubes 0, 1 and 2, there were sometimes no flies at all. Therefore, we decided to pool flies from different replicates of the first session that had made the same number of choices to the light, as we have done for the t-maze experiments (see above). We performed a total of 139 first sessions to reach at least 40 flies for each tube for the second session. After performing the two session experiments, we randomly selected 8 replicates from each tube for analysis, except for tube 0 for which we only managed to obtain four replicates. The tubes were cleaned thoroughly after each test.

## Buridan

On the first day, a group of flies was split up according to their phototactic preference in CPP, and samples from each tube were prepared to assess their locomotion in Buridan's paradigm [45]. Briefly, after completing a session in the CPP, flies from each tube were selected and had their wings shortened under cold anesthesia (protocol DOI: 10.17504/protocols.io.c7vzn5). They were left to recover overnight within individual containers, with access to water and sugar (local store) before being transferred to the experimental setup. The setup consists of a round platform (117 mm in diameter) surrounded by a water-filled moat placed at the bottom of a uniformly illuminated white cylinder (313 mm in height) with 2 stripes of black cardboard (30 mm wide, 313 mm high and 1 mm thick) placed 148.5 cm from the platform center, opposite each other. The experiment duration was set to 900 s. As explained before, after a first session of choices in CPP it was difficult to find flies in some of the tubes. Therefore, we performed as many CPP experiments as required in order to have between 13 and 15 flies from each tube to be tested in Buridan's paradigm. Whenever more than one fly was present in a tube (commonly occurring for tubes 3, 4 and 5) we randomly picked the fly to be tested. All the samples were drawn from multiple experiments, none from a single experiment. Data were analyzed using BuriTrack and CeTrAn [45] (RRID:SCR_006331), both available at http://buridan.sourceforge.net. We considered the following six behavioral parameters.

**Activity time per minute.** We considered every movement as activity and every absence of movement lasting longer than 1 s as a pause (shorter periods of rest were considered as active periods). Then, we calculated for each fly the total activity time (in seconds) and normalized it to the total time of the experiment.

**Pauses per minute.** An absence of movement for periods longer than 1s were considered as a pause. Then, the total number of pauses in the experiment were normalized to the total time of the experiment.

**Distance traveled.** The total distance traveled by each fly was calculated by adding up every movement length over the whole experiment.

**Median speed.** First, we calculated the instant speed for each movement (in mm/s). We divided the distance traveled by the time (always 0.1 s with 10 Hz sampling rate). We then report the median speed for each fly. Speeds exceeding 50 mm/s were considered to be jumps and were not included in the median speed calculation.

**Pause duration.** The median duration of pauses was calculated for each fly.

**Number of walks per minute.** This value was calculated as the total number of times the fly walked from one stripe to the other in the experiment, normalized to the total time of the experiment.

**Stripe deviation.** This metric corresponds to the angle between the velocity vector and a vector pointing from the fly position toward the center of the frontal stripe. For each displacement, the vectors going from the fly position toward both stripes are calculated and the respective angles between the velocity vector and each of those vectors are measured. The smaller of the two angles is chosen as output (corresponding to the angle with the stripe most in front of the animal). The median of all deviation angles was reported for each fly (in degrees).

**Meander.** This value is a measure of the tortuosity of the trajectories. It was calculated by dividing the turning angle, i.e., the angle between two consecutive velocity vectors, by the instantaneous speed. The median was calculated for each fly (in degrees×s/mm).

## Statistical analysis

Statistical analyses were performed with INFOSTAT, version 2013 (Grupo InfoStat, Facultad de Ciencias Agropecuarias, Universidad Nacional de Córdoba, Córdoba, Argentina) and R

(http://www.r-project.org/). As dictated by the experimental design and data composition Kruskal-Wallis followed by Rank-test was performed. Normality was tested using the Shapiro–Wilks test, and the homogeneity of variance was assessed with Levene's test. Following the arguments in [47], to avoid false positives, a value of p<0.005 was considered statistically significant.

Principal component analysis (PCA) in Fig 5 was based on correlation matrices, and the minimum eigenvalue was set at 1. PCA reduces the original variables to a smaller number of new variables called principal components (PCs). The PCA output includes the PCs scores for the samples introduced in the analysis, the eigenvectors, and the loadings.

## Results

### Fly behavior is not consistent between two consecutive tests

In order to determine the ideal number of flies to be tested as a group in the T-Maze in one session, we evaluated the photopreference (Choice Index -CI-) of groups from 20 to 100 individuals. This index was not significantly different for the selected groups of flies (Fig 2A; Kruskal-Wallis, H = 14.27, p = 0.006; Rank test). The variability of the Choice Index was significantly higher for the group of 20 flies compared with all the other groups (variance: 20 = 0.16, 40 = 0.03, 60 = 0.04, 80 = 0.01, 100 = 0.02, Levene's test: F = 4.69, p = 0.004; Tukey's test p<0.05). A group size of 80 flies sported a set of attractive features: it was the least variable group, the compartments of the T-Maze were far from overcrowded, and after the experiment there were still enough flies to be meaningfully split into groups. Therefore, for all subsequent T-maze experiments, we used groups of 80 flies.

We measured the amount of variability in the overall photopreference by measuring the same groups of 80 flies over a four-day time course. No significant differences in CI over the four experiments were observed (Fig 2B; Kruskal-Wallis, H = 6.49, p = 0.09).

With these results showing that overall choices were consistent within groups of 80 flies over several days, we tested whether a previous light/dark choice would influence a subsequent choice. For this test, we separately re-tested the subgroups of flies which chose the light or the dark arm of the T-maze in a first session of the experiment, respectively. Because the number of flies in the dark arm was much lower than that in the bright arm, we added the flies found in the (dark) elevator and randomly combined Dark+Elevator subgroups from different replicates (see Methods for details). Consistent with the original observation mentioned by S. Benzer (cited by [39]), we observed a similar distribution of the flies in the T-maze when retesting in a second session the photopositive and photonegative subgroups 3 h after their first choice (Fig 2C; Kruskal-Wallis, H = 2.64, p = 0.267). In contrast, when the flies were re-tested 24 h after the first test, we observed a lower Choice Index for the photonegative group of flies compared to the Original group and the Bright subgroup (Fig 2D; Kruskal-Wallis, H = 18.7, p = 0.0001). However, as the CI is positive, even 24h later, most flies that chose the dark arm on the first day chose the bright arm on the second day.

### Fly individuality appears with more choices

As these results suggested that flies do not behave consistently, we chose a different experiment, where increasing the number of consecutive tests was technically less challenging. In the classic CPP designed by Seymour Benzer [42], the flies separated according to five consecutive choices in a single session. Each choice in this experiment involves either staying put or walking towards the light (positive phototaxis). The consequence of one such session with 5 consecutive choices is that the original group is split into six subgroups according to their sequence

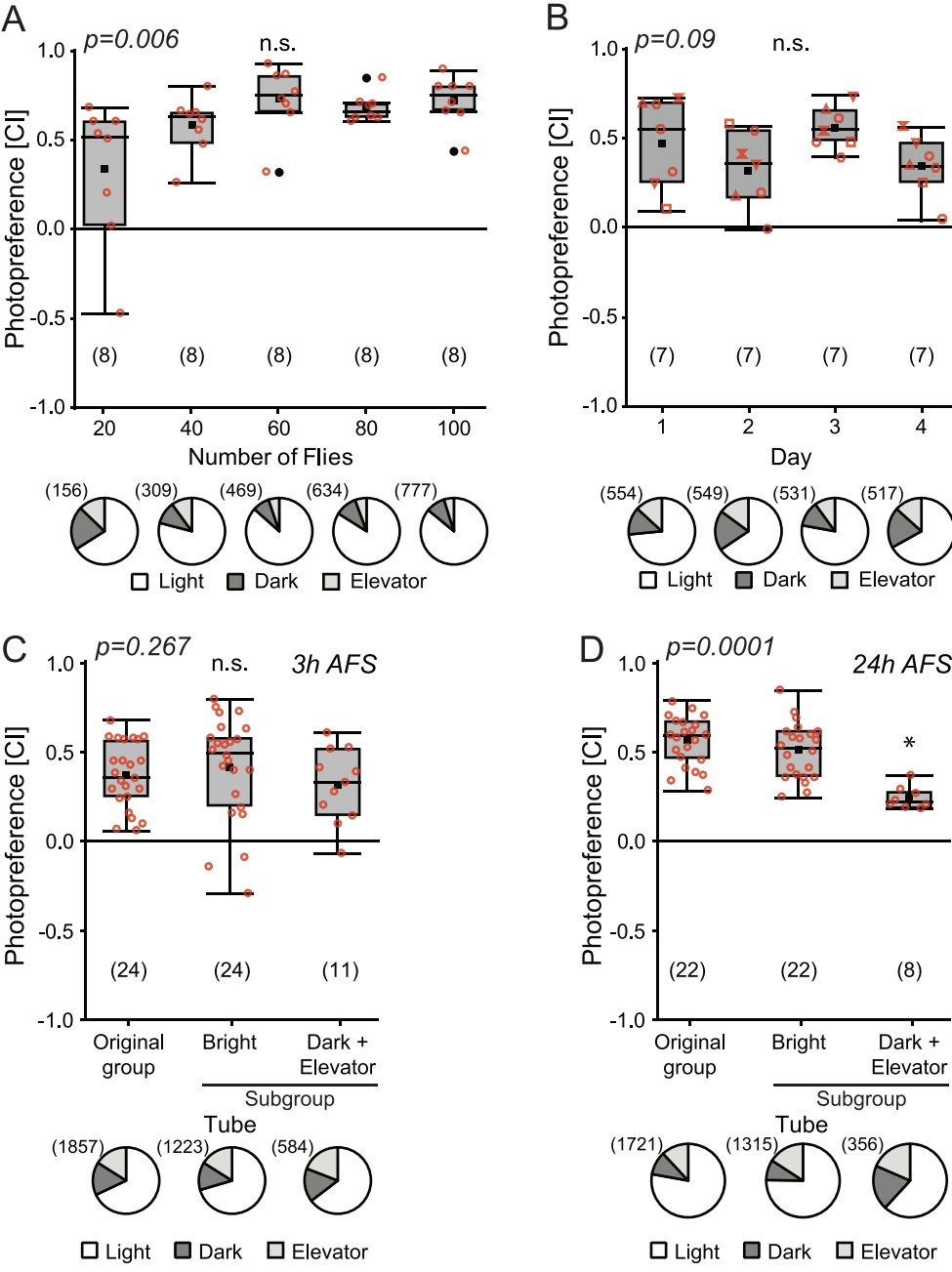

**Fig 2. Photopreference results are independent of prior testing. (A)** Groups of 20, 40, 60, 80 or 100 flies were tested for their photopreference in a light/dark T-maze. Upper: Box plots depicting photopreference distributions. Lower: Pie charts illustrate the distribution of all tested flies in each compartment of the T-Maze **(B)** Groups of 80 flies tested for photopreference on consecutive days. **(C and D)** Photopreference of subgroups generated after the first photopreference session (AFS). (C) 3h After the first test. (D) 24h after the first test. Box plots denote quantiles 0.05, 0.25, 0.75 and 0.95, median, mean (black square), and outliers (black dots). Individual data points are shown in red circles. Same shapes in B represent individual data points from different experiments over time. * p<0.005. n.s: not significant. Numbers in brackets represent sample size.

of choices: the flies choosing five times to walk towards the light end up in tube 5, while the ones never walking remain in tube 0.

As before, we first studied the impact of the number of flies per session and of the number of repeated sessions. There were no significant differences in phototaxis index (PI) for the

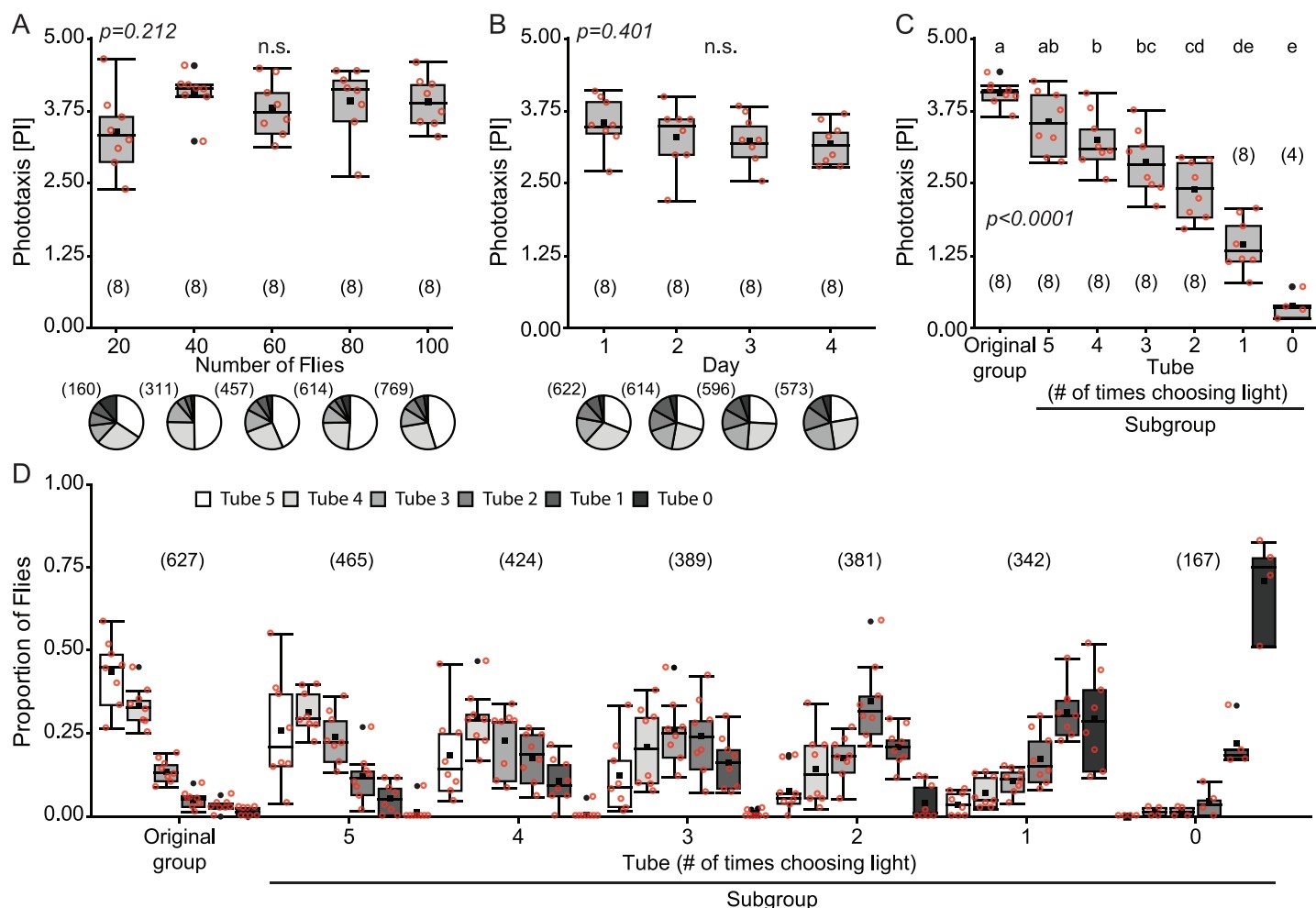

**Fig 3. Repeated testing reveals individuality of flies also in group tests.** **(A)** Groups of 20, 40, 60, 80 or 100 flies were tested for phototaxis in the CPP. Upper: Box plots of phototaxis index (PI). Lower: Pie charts illustrate the distribution of all tested flies in each tube. **(B)** Repeated phototaxis tests of flies in groups of 80 individuals on four consecutive days. **(C)** Phototaxis of subgroups generated by a single prior session CPP (Original). p<0.005: different letters indicate significant differences. **(D)** Proportion of flies in each tube in the experiment depicted in (C). Box plots denote quantiles 0.05, 0.25, 0.75 and 0.95, median, mean (black square), and outliers (black dots). Individual data points are shown in red circles. n.s: not significant. Numbers in brackets indicate sample size.

number of flies tested, although the values of the group with 20 flies were slightly lower and more variable than the rest of the groups (Fig 3A; Kruskal-Wallis, H = 5.84, p = 0.212). For the next phototaxis experiments we tested 80 flies per group, in order to keep it as similar as possible to the T-Maze experiments.

When the same group of flies was tested in the CPP over the course of four consecutive days, the PI was not significantly affected (Fig 3B; Kruskal-Wallis, H = 2.94, p = 0.401).

As previously mentioned, each session of a CPP has five consecutive choices. To double the number of choices from five to ten, we took each of the six subgroups of flies generated after a first session of CPP and performed a second session to each one separately on the next day. Analogous to the T-maze experiment, we pooled flies from several first sessions of the CPP experiments in order to achieve a sufficient number of flies in the second session (see Methods). We observed a strong correlation between the first set of choices (first session) of the flies and the choices made on the re-test (second session). The fewer times flies chose to walk towards the light in the first session, the lower the PI for that group on the re-test (Fig 3C;

Kruskal-Wallis, H = 41.2, p<0.0001). Analogous to other countercurrent designs, also here, this countercurrent machine effectively fractionates an initial group of flies into groups of less and less photopositive individuals (Fig 3D), approaching the results described in [38].

## General activity parameters modulate phototaxis

In contrast to the T-maze, where each choice involves the same general tasks, the choices in the CPP are asymmetrical: one option involves walking, the other involves staying. Therefore, general differences between the flies such as walking speed or inclination to walk may confound the results when the flies have been fractionated in ten consecutive choices. To estimate the extent to which such orthogonal inter-individual differences contributed to the differences observed in the CPP experiments, we evaluated individual flies from each of the six tubes after their first session in the CPP in Buridan's Paradigm, where flies walk between two unreachable opposing black landmarks (vertical stripes; [44,45]. We found significant differences between the six tube-samples in five out of the eight parameters we evaluated (see Methods). Flies less likely to show positive phototaxis travelled less total distance (Fig 4A; Kruskal-Wallis, H = 29.0, p<0.0001), completed fewer walks between the landmarks (Fig 4B; Kruskal-Wallis, H = 24.1, p = 0.0002), showed a lower median duration of activity (Fig 4C; Kruskal-Wallis, H = 18.1, p = 0.0028), and a decreased median speed (Fig 4D; Kruskal-Wallis, H = 19.6, p = 0.0015). Moreover, we found significant differences in meander values (Fig 4E, H = 19.19, p = 0.0018), indicating that flies that show strong phototaxis tend to walk straighter in Buridan's paradigm, compared to flies showing weaker phototaxis in the CPP. In contrast, the pauses per min and the duration of the pauses were not significantly different among the groups (Fig 4F, Kruskal-Wallis, H = 3.67, p = 0.5971, and 4G, Kruskal-Wallis, H = 3.19, p = 0.6701). Also, landmark fixation (stripe deviation) was not significantly different across tubes (Fig 4H, H = 1.85, p = 0.87).

To better understand the association between the CPP results and Buridan's variables, we carried out a standardized principal component analysis (PCA) including the number of light choices (i.e. tube where the fly ended after a first session of CPP) and the eight activity measures as variables. The number of light choices had a higher contribution to PC1 (value of loading = 0.61) than PC2 (value of loading = 0.17) or PC3 (value of loading = -0.35). Interestingly, the variables that were significant between flies from different tubes after a first session of CPP in our previous analysis, have also higher contributions to PC1. The biplot diagram of PC1 vs. PC2 shows that this phototaxis variable is highly positively correlated with distance traveled, as the two vectors share almost the same orientation and sense (Fig 5). This is in accordance with the significant decrease in distance traveled according to the tube (Fig 4A). Similar results are observed for the positive correlation of walks per min, and the negative correlation of meander with the number of light choices (Fig 4B and 4E).

## Discussion

### Individuality and spontaneity interact

We found that after a single choice between a bright and a dark arm in a T-maze, flies distributed in the same manner in a second choice, regardless of what their first choice was (Fig 2C and 2D), reproducing the spontaneous choice behavior reported previously [39,40] and which indicated that current choices were independent of previous choices (chance). Increasing the number of choices from two to ten using Benzer's counter-current apparatus, it emerged that flies which showed a strong preference in five consecutive choices were biased towards the same preference also in subsequent choices (Fig 2C and 2D), evincing a degree of individuality in such choices similar to that found in experiments with single flies [38] These results indicate

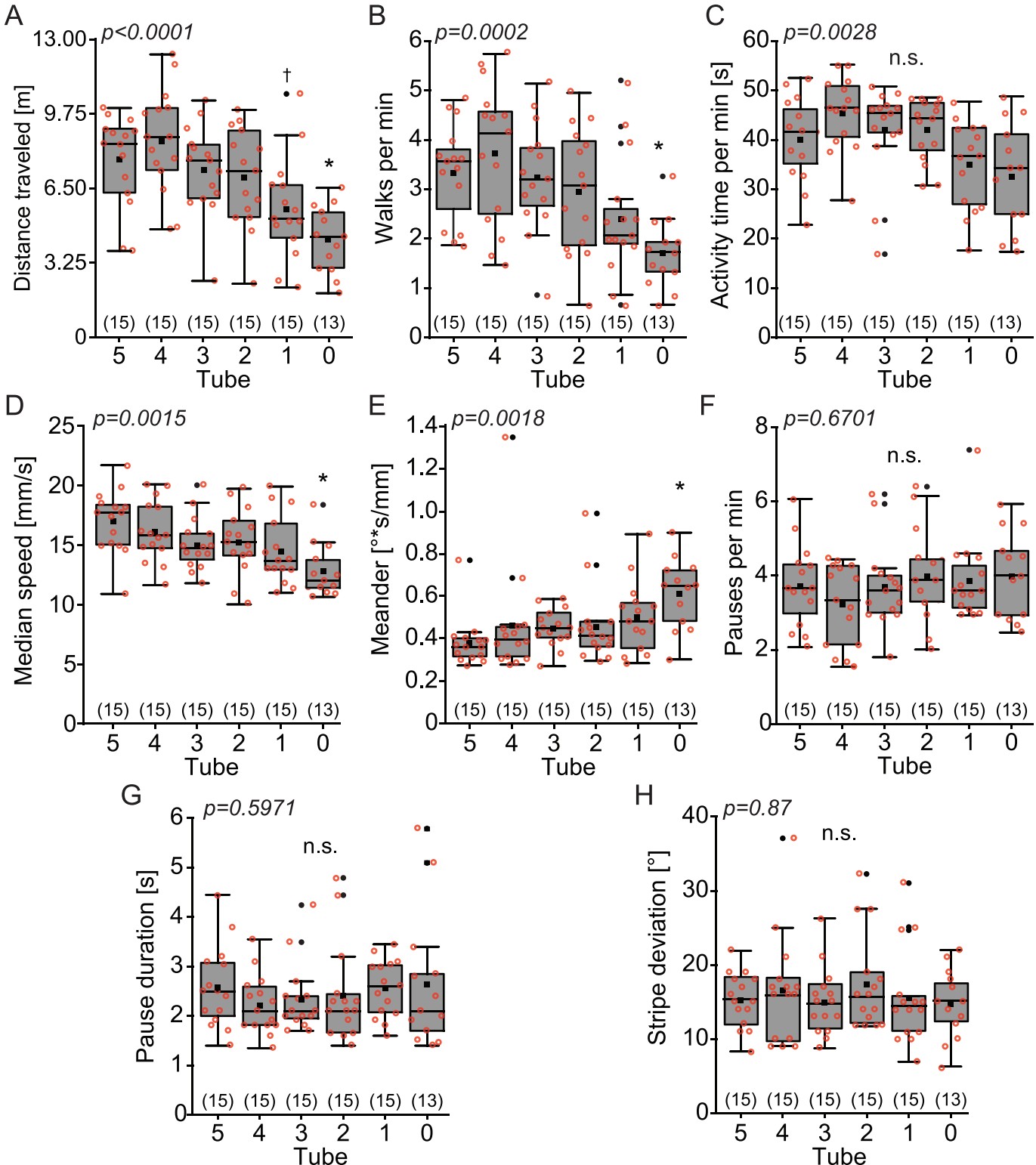

**Fig 4. Less phototactic flies show reduced activity in Buridan's paradigm.** Groups of 80 flies were tested in Buridan's paradigm after they were fractionated according to their phototaxis in CPP. Higher tube number indicates stronger phototaxis. **(A)** Total distance travelled during the experiment. p<0.005 with. * compared with tubes 5, 4, and 3; † compared with tube 4. **(B)** Number of walks between the stripes. p<0.005 with * compared with tubes 5, 4, and 3. **(C)** Activity time per minute. Kruskal-Wallis significant, p<0.005, Rank test not significant. **(D)** Median walking speed. p<0.005 with * compared with tube 5. **(E)** Meander, a measure for the tortuosity of the fly's trajectory. p<0.005 with * compared with tube 5. **(F)** Pauses per minute. Not significant. **(G)** Pause duration. Not

significant. **(H)** Stripe deviation, a measure of landmark fixation (lower values indicate better landmark fixation). Not significant. Box plots denote quantiles 0.05, 0.25, 0.75 and 0.95, median, mean (black square), and outliers (dots). Individual data points are shown in red circles. n.s: not significant. Numbers in brackets indicate sample size, p-values on top of each panel indicate Kruskal-Wallis p-values.

that current choices show some dependence on previous choices (necessity), even when the flies are tested in groups. Flies do not alter their photopreference behavior when tested in groups compared to experiments with single flies. Thus, what appeared, at first, as a contradiction, can be explained by a combination of chance and necessity, with two different

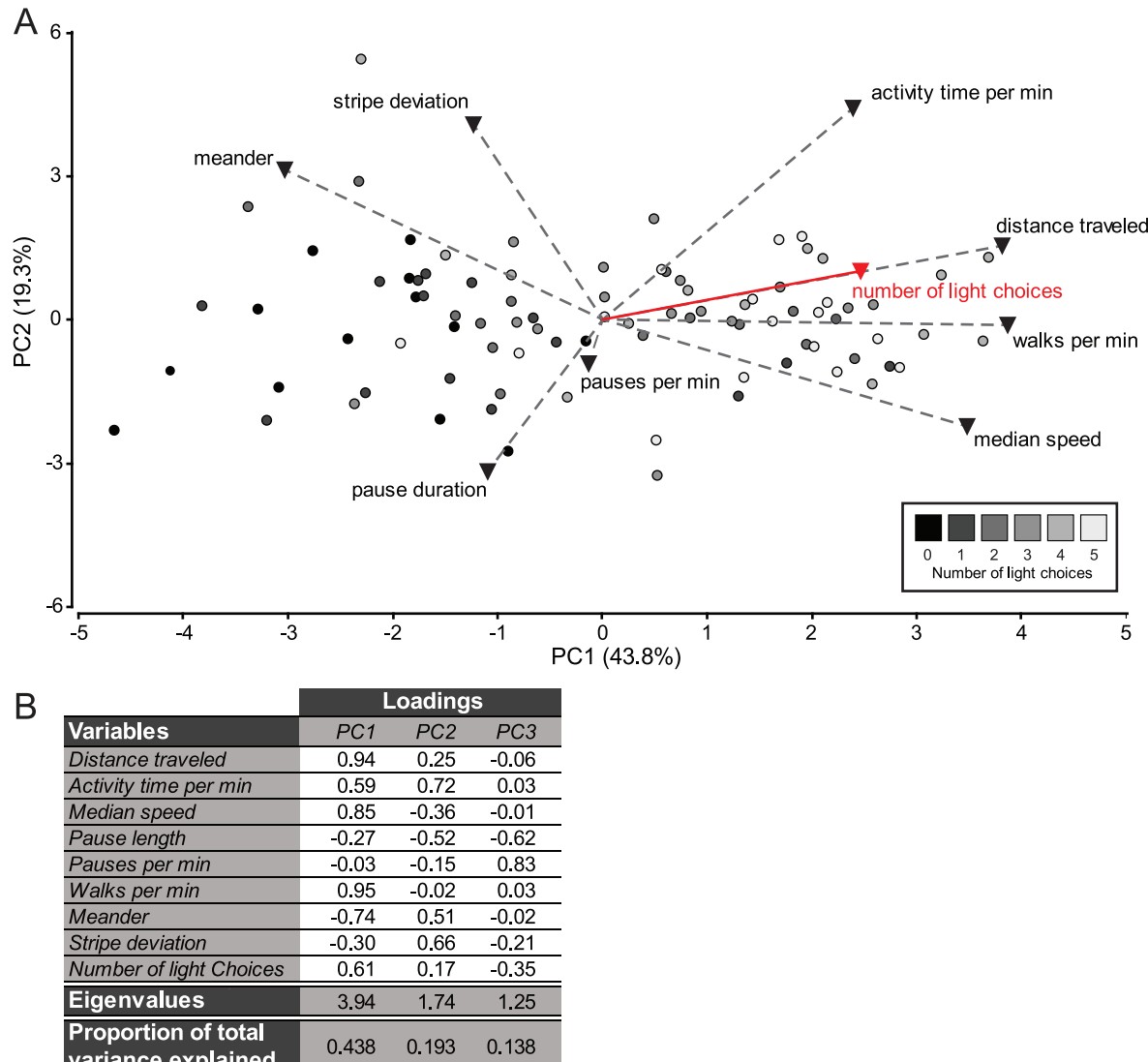

**Fig 5. The number of light choices is correlated with significant activity measures. (A)** PCA biplot diagram of Principal Component 1 (PC1) vs. Principal Component 2 (PC2). Samples are displayed as dots of different colors depending on their number of light choices. Variables are displayed as vectors: activity measures, dashed lines with black triangles; number of light choices, red line and triangle. The length of a vector is proportional to the variance of the corresponding variable. The angle between the vectors approximates the correlation between the variables they represent. The numbers in brackets represent the amount of variability in the original variables explained by the principal component. (B) Loadings, Eigenvalues and the Proportion of Variance explained for each of the first three Principal Components. The loadings of the PCs are the correlations of each variable on each PC. Therefore, original variables with high loadings (either positive or negative) are more closely related to the patterns revealed by the PC.

mechanisms contributing to behavioral variability. In single choices, spontaneity obscures individuality even in phototaxis.

On the level of the individual, there are few documented examples of rigid stereotypy (likely, because predictability is not an evolutionary stable strategy, see [11,48–54] for examples) and many examples of adaptive behavioral variability [11,55–61]. The variability we found here in photopreference is consistent with even this behavior, often described as iconically rigid, being more accurately characterized by flexible decision-making components, rather than simply constituting a stereotypic response [43]. Due to the inherent probabilistic nature of such behaviors [27], any individuality modulating to the individual's choices will manifest itself only after a sufficient number of choices have been observed (Fig 3 and [38]). That being said, even after selecting for extremely phototactic animals, the selected groups still contained individuals making consecutive choices in the opposite direction (Fig 3D). Taken together, these results suggest that this is a classic case of spontaneity making the behavior less consistent than it would be if sensory input were the sole determinant of the behavior. There are first results from other animals which suggest ongoing, intrinsic neural activity may be contributing to such processes [61]. These corroborate the notion that adaptive behavioral choice is a nonlinear dynamic process that is influenced by its own history (i.e., a process with 'memory'). Renewal processes without memory (e.g., Markov, Lévy or Cox processes) have already proven inadequate to model fly behavior before [27]. Our results here are therefore consistent with the hypothesis that probabilistic behavior does not arise from pure stochasticity but rather that stochasticity is harnessed within allostatic, non-ergodic, nonlinear neural networks.

On the level of the group, a second probabilistic aspect comes to play, that of the tested sample. If a population consists of individuals, each with a different average preference (see previous paragraph), the observed preference of the sample will reflect the mean of these average preferences. Group behavior is thus best characterized by an averaging of averages, yielding little insight into individual decision-making mechanisms other than general biases that exist in the population, but that manifest themselves on the individual level to an unpredictable degree.

The combination of chance (spontaneity) and necessity (individuality) described here can serve as a particularly ostensive example why even very high predictability on the level of group averages so often fails to predict individual behavior. Even though a population mean may indicate that the tested flies are overwhelmingly photopositive, this does not entail that these flies must be photopositive when tested repeatedly, let alone that any single fly will behave photopositively in any given choice. The fact that this is the case even for our simple experiments using iconically stereotypic behaviors in animals with numerically less complex nervous systems, can serve as a stark reminder of how little information population averages carry for any human individual belonging to any sample tested for, e.g., any cognitive trait.

## The shoulders of giants

The literature on insect behavior contains many observations that group experiments are representative of multiple individual experiments [40,42]. Some of these observations go back decades and have since become established tenets of behavioral science, despite rarely, if ever, being accompanied by experimental data. With some scholarly fields suffering from severe problems replicating other established tenets of science, testing the shoulders on which we stand has perhaps never been more important. Here, we have tested photopreference in the fruit fly *Drosophila* in two classic test situations, Benzer's counter-current apparatus and a T-maze. Both sets of experiments successfully replicate previous results, and their combination

resolves the apparent contradiction in the literature (see above). At the same time, these experiments rule out the option that in photopreference, fly behavior changes when they are tested in groups, compared to single animal experiments.

## General contributions to individuality in phototaxis

Even in such simple experiments as the ones described here, the mechanisms underlying individuality are numerous. We find that individuals which showed little positive phototaxis in the counter-current apparatus also had lower activity levels in Buridan's paradigm (Fig 4). These results suggest that part of this difference may not be exclusively due to differences in the internal light/dark preference but also in general walking activity. Separate experiments need to investigate which components are genetic and which rely on other mechanisms and their relative contributions to individuality. Our results also emphasize the need for carefully designed experiments capable of isolating the components of individuality, if the biological mechanisms underlying it are to be discovered.

## Acknowledgments

We thank Academic Editor Dr. Efthimios M. C. Skoulakis, and the anonymous reviewers for their suggestions and comments.

## Author Contributions

**Conceptualization:** Björn Brembs, E. Axel Gorostiza.

**Data curation:** E. Axel Gorostiza.

**Formal analysis:** Luciana M. Pujol-Lereis, E. Axel Gorostiza.

**Funding acquisition:** Björn Brembs.

**Investigation:** Isabelle Steymans.

**Supervision:** Björn Brembs, E. Axel Gorostiza.

**Writing – original draft:** Luciana M. Pujol-Lereis, Björn Brembs, E. Axel Gorostiza.

**Writing – review & editing:** Björn Brembs, E. Axel Gorostiza.

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
