## [Decision Letter · Decision Letter 0]

11 May 2021

PONE-D-21-09940

Collective action or individual choice: Spontaneity and individuality contribute to decision-making in Drosophila

PLOS ONE

Dear Dr. Gorostiza,

Thank you for submitting your manuscript to PLOS ONE. After careful consideration, we feel that it has merit but does not fully meet PLOS ONE’s publication criteria as it currently stands. Therefore, we invite you to submit a revised version of the manuscript that addresses the points raised during the review process.

Although both reviewers and myself concur that the manuscript has merit, there are still a number of clarifications, both technical and semantic that need to be thoroughly addressed. Please make sure the methods and statistics are thoroughly explained.

We look forward to receiving your revised manuscript.

Kind regards,

Efthimios M. C. Skoulakis, PhD

Academic Editor

PLOS ONE

Journal Requirements:

Reviewers' comments:

Reviewer's Responses to Questions

**Comments to the Author**

1. Is the manuscript technically sound, and do the data support the conclusions?

Reviewer #1: Yes

Reviewer #2: Partly

2. Has the statistical analysis been performed appropriately and rigorously? 

Reviewer #1: Yes

Reviewer #2: I Don't Know

3. Have the authors made all data underlying the findings in their manuscript fully available?

Reviewer #1: Yes

Reviewer #2: Yes

4. Is the manuscript presented in an intelligible fashion and written in standard English?

Reviewer #1: Yes

Reviewer #2: No

5. Review Comments to the Author

Reviewer #1: In this manuscript, Steymans, et al. examined the consistency of fly to choose to move into a lighted chamber in two different, but related, phototaxis experimental designs. Wild type Drosophila melanogaster are strongly positively phototactic. The authors reexamined the issue of consistency in choice of moving to a lighted chamber by resampling the choices of flies after they have moved into a lighted chamber or remained in a darkened one. The question is interesting as it directly relates to the interpretations of a fly’s behavior in a T-Maze or counter current apparatus, and it sheds light on why some behaviors are found to be more consistent when assayed in groups of flies rather than individually.

Previous work by Seymore Benzer found that the distribution of flies in a phototaxis experiment appeared to be probabilistic; if 70% of flies chose the lighted chamber, a resampling of the flies would show the same distribution regardless of whether the initially choose the lighted chamber or the darkened chamber. However, the conservation of this distribution was not found by work reported in Kain et al, 2012, in which individual flies were repeatedly scored for movement towards light in a very different apparatus. In Kain, et al., the choice to move towards the light varied between individuals.

In Steymans, et al., the authors initial findings from resampling in the T-maze and counter-current apparatus more closely resembled that of Benzer et al. However, after increasing the number of behavior choices in the counter-current apparatus, the authors found that the previous choice strongly correlated with subsequent choices, similar to Kain et al’s finding. The authors offer the hypothesis that this emergence of consistency in choice that emerged in this retesting and in Kain’s et al. experiments is due in part to the inability to assess individual choices in group behavior in the initial counter-current experiments. The difficulty is assessing the consistency (referred to be the authors as Necessity or the need to behavior in a certain way), is also complicated by the presence of spontaneity in an animal’s behavioral choices. These are reasonable inferences.

Major Comments:

When the fly is in the counter-current apparatus or the T-maze, the flies are normally free to move back and forth. The selection they “make” is dictated by their position when the trial ends (30 seconds for T-maze and 15 seconds for the counter-current Apparatus). These are relatively short periods of time for the animals to explore and make a stable choice. The authors should discuss the choice of these time points and how they may impact the measurements of individual choice in phototactic preference.

The Buridan’s paradigm controls are an important part of this paper as they provide a mechanistic understanding for why some flies show lower phototactic behavior in the counter-current apparatus. In these experiments, it would be good if the authors reported Principle Component Analysis, to see how differences in phototaxis load with the same activity measures.

Minor Comments:

In the abstract, the authors refer to flies with an “extremely negative first choice”. Since the choices are binary, I do not understand how a negative choice cab be extremely, moderately or otherwise.

The first sentence of the Introduction mentions a Hallmark of success neuroscience is the ability to predict behavior. Although I understand this helps frame the author’s question, I know of few neuroscientists that would agree with this criterion. Perhaps this can be rephrased so as not to appear to be a straw man.

Reviewer #2: The paper nicely addresses two sets of seemingly canonical and conceptually most influential experiments on the consistency/ variability of behavior that are talked about a lot, but hardly ever substantiated by actual data.

I refrain from reporting in my own words what the authors did. I just report my main concerns to be i) the Methods are not explained with sufficient clarity; to some extent, good info graphics may help; ii) the possible contribution of learning processes is not properly considered.

The “or” in the title implies a dichotomy of action vs choice that the authors may not actually want to be implied. Please reconsider the title.

The Introduction features a number of (words for) behavioural paradigms that are not properly defined or explained. Such as “phototaxis, the paradigm mentioned in the Benzer-quote, the one used by Quinn et al, light-dark T-maze, classic countercurrent phototaxis experiment, Benzer phototaxis experiments”.

Please clearly define these terms and do not use synonyms, and apply the defined terms consistently throughout the manuscript.

Were animals from both sexes used? If not, why not? and: which sex was used? If both sexes were used: could you separate the data by sex?

Not all terms of the CI equation are defined, and FE is defined in the text but not used in the equation. Please check.

Line 172: Better something like “We added the flies from the elevator to the ones in the dark for retesting…?

Line 174: I do not quite understand what you did (and hence the rational) of the procedure described as “Second, …”.

Line 223, after “Therefore…”: I do not get exactly what was done here. Partially, this might be because of uncertainty as to what is the meaning of “repetition, session, choice, experiment, replicate, test” in the preceding and following sections.

Line 289: Do you imply the flies indeed jumped, or do you consider these to be cases of artefact from tracking?

In general, carefully designed info-graphics for the behavioural paradigms would seem very useful.

The Benzer paradigm is explained much better in the Results than in the Methods section…

Line 237: Please explain why what appears to be a very conservative P-threshold was used (P< 0.005(!)).

Line 373: the wording “… we tested whether a previous light/dark choice would influence a subsequent choice” implies some sort of learning, and indeed it would seem important to consider such learning throughout the manuscript as a possibility (also @Introduction). One could view such learning as a “personalizing” of chance/ of action probability fields, really. This might be what happens when, as the authors put it in a subheading “Fly individuality appears with more choices”.

In other words, the flies may behave according to a given probability of choosing dark, and that probability can be changed by the experience the flies make based on this very choice (or multiple such choices). This would call into question the dichotomy the authors put up: for a given fly the behavior is according to its own, self-tailored and adjustable probability. In yet other words, each fly throws a dice as to what to do and does so each time anew, but it throws “its own” dice, one that it has shaped itself before and one that changes at least slightly every time actions were based on what the dice returned.

Interestingly, in the work of Pamir et al 2014 Front Behav Neurosci on bees, the animals are suggested to learn in an all-or-none manner, so to speak switching from one rigidity (not-showing PE towards an odour) to another rigidity (showing PE).

I wonder whether canonical learning mutants would be less well “selectable” than the ones in Figure 2…?

Line 439: I cannot quite certainly follow what was done here; this means I did not get the difference/ relationship between Figures 2C versus D. This would seem essential to judge a key conclusion, though.

If I understood correctly, it might help to label the X-axis with “Number of times choosing light” or so (I hope I got the sign right here…). Using the abstract expressions “Tube #1, …” would be intuitive only with the suitable graphic/ pictogram.

Maybe arrange Figure 3 such that the significantly different data are grouped together?

Maybe the data from figure 3 would allow some sort of machine-learning approach to see how many “kinds of flies” there are?

Line 569: It should be “than” and not “then” here.

6. PLOS authors have the option to publish the peer review history of their article (what does this mean?). If published, this will include your full peer review and any attached files.

Reviewer #1: No

Reviewer #2: No

---

## [Author Response · Author response to Decision Letter 0]

27 Jul 2021

Dear Dr. Skoulakis and Reviewers,

It is my great pleasure to submit a revised and improved version of the research article entitled “Collective action or individual choice: Spontaneity and individuality contribute to decision-making in Drosophila”.

We would like to thank you and the reviewers for taking the time for carefully reading and analyzing the manuscript. We are pretty confident in saying that your comments certainly prompted a better version of this manuscript. We have addressed the recommendations of both reviewers and have made the following changes:

Reviewer #1

- When the fly is in the counter-current apparatus or the T-maze, the flies are normally free to move back and forth. The selection they “make” is dictated by their position when the trial ends (30 seconds for T-maze and 15 seconds for the counter-current Apparatus). These are relatively short periods of time for the animals to explore and make a stable choice. The authors should discuss the choice of these time points and how they may impact the measurements of individual choice in phototactic preference.

Answer: This is a very good point. We chose 30s and 15s, respectively, based on previous observations, experiments, and certain criteria that are now explained in methods section. Briefly, the time was adjusted based on the length of the tubes and the distance a fly should travel if they wanted to reach an end and switch the preference in that time. According to our previous Buridan experiments, the average speed of the slowest flies in our hands is around 1.5 cm/sec. In both paradigms, flies could travel in the given time at least twice the distance from a tube, i.e. the flies could choose a side, get to the end of that side and go back to the other side. 

Moreover, in one of our previous papers we gave the flies 3min to choose between Darkness and Light in the same T-mazes and their choices were not different from the ones obtained for flies with only 30s to choose.

- The Buridan’s paradigm controls are an important part of this paper as they provide a mechanistic understanding for why some flies show lower phototactic behavior in the counter-current apparatus. In these experiments, it would be good if the authors reported Principle Component Analysis, to see how differences in phototaxis load with the same activity measures.

Answer: A Principal Component Analysis was performed and a new Figure 5 was generated. 

- In the abstract, the authors refer to flies with an “extremely negative first choice”. Since the choices are binary, I do not understand how a negative choice cab be extremely, moderately or otherwise.

Answer: This was a mistake, and we corrected it.

- The first sentence of the Introduction mentions a Hallmark of success neuroscience is the ability to predict behavior. Although I understand this helps frame the author’s question, I know of few neuroscientists that would agree with this criterion. Perhaps this can be rephrased so as not to appear to be a straw man.

Answer: References were added to support our statement.

Reviewer #2

- I refrain from reporting in my own words what the authors did. I just report my main concerns to be i) the Methods are not explained with sufficient clarity; to some extent, good info graphics may help; ii) the possible contribution of learning processes is not properly considered.

Answer: We would like to apologize for the lack of clarity. After these comments we realized that the manuscript was too confusing. We tried to address all suggestions on this matter, like avoiding synonyms. We added a new figure to explain the paradigms and the procedures, and used consistent nomenclature for the different aspects of the experiments. We have also worked on all other comments and suggestions. 

- The “or” in the title implies a dichotomy of action vs choice that the authors may not actually want to be implied. Please reconsider the title.

Answer: Our original intention was to draw attention to this dichotomy, which has been around for many years. In our manuscript we showed that it is not only an individual choice but also spontaneity in the behavior that contributes to final action, and therefore, even studying highly predictable group behaviors are not good to understand individual decisions.

- The Introduction features a number of (words for) behavioural paradigms that are not properly defined or explained. Such as “phototaxis, the paradigm mentioned in the Benzer-quote, the one used by Quinn et al, light-dark T-maze, classic countercurrent phototaxis experiment, Benzer phototaxis experiments”.

Please clearly define these terms and do not use synonyms, and apply the defined terms consistently throughout the manuscript.

Answer: As mentioned before, we realized our use of the terms was confusing and we made the corrections required.

- Were animals from both sexes used? If not, why not? and: which sex was used? If both sexes were used: could you separate the data by sex?

Answer: An explanation was added in the manuscript. Briefly, we know from previous experiments published and unpublished that in these paradigms females and males behave indistinguishable. Therefore, we randomly picked flies regardless of their sex. 

- Not all terms of the CI equation are defined, and FE is defined in the text but not used in the equation. Please check.

Answer: The information was added.

- Line 172: Better something like “We added the flies from the elevator to the ones in the dark for retesting…?

Answer: The suggestion was introduced in the text 

- Line 174: I do not quite understand what you did (and hence the rational) of the procedure described as “Second, …”.

Answer: The text was rephrased to clarify the protocol, and a new figure was also added for a better explanation. 

- Line 223, after “Therefore…”: I do not get exactly what was done here. Partially, this might be because of uncertainty as to what is the meaning of “repetition, session, choice, experiment, replicate, test” in the preceding and following sections.

Answer: As mentioned before, we realized our use of the terms was confusing, and we made the corrections needed.

- Line 289: Do you imply the flies indeed jumped, or do you consider these to be cases of artefact from tracking?

Answer: Flies actually jumped and it is possible to see this in the videos or in the traces as extremely fast and straight displacements of the flies. 

- In general, carefully designed info-graphics for the behavioural paradigms would seem very useful.

Answer: As mentioned before, we added a new figure explaining the paradigms and experimental procedures.

- The Benzer paradigm is explained much better in the Results than in the Methods section…

Answer: Thank you for noticing this. We copied that explanation into the method section.

- Line 237: Please explain why what appears to be a very conservative P-threshold was used (P< 0.005(!)).

Answer: We took this decision to avoid false positives. Reference was included in the manuscript.

- Line 373: the wording “… we tested whether a previous light/dark choice would influence a subsequent choice” implies some sort of learning, and indeed it would seem important to consider such learning throughout the manuscript as a possibility (also @Introduction). One could view such learning as a “personalizing” of chance/ of action probability fields, really. This might be what happens when, as the authors put it in a subheading “Fly individuality appears with more choices”.

In other words, the flies may behave according to a given probability of choosing dark, and that probability can be changed by the experience the flies make based on this very choice (or multiple such choices). This would call into question the dichotomy the authors put up: for a given fly the behavior is according to its own, self-tailored and adjustable probability. In yet other words, each fly throws a dice as to what to do and does so each time anew, but it throws “its own” dice, one that it has shaped itself before and one that changes at least slightly every time actions were based on what the dice returned.

Answer: Comments related to this topic were added into the manuscript.

- Interestingly, in the work of Pamir et al 2014 Front Behav Neurosci on bees, the animals are suggested to learn in an all-or-none manner, so to speak switching from one rigidity (not-showing PE towards an odour) to another rigidity (showing PE).

Answer: This and other manuscripts related are now mentioned in the manuscript.

- I wonder whether canonical learning mutants would be less well “selectable” than the ones in Figure 2…?

Answer: This is a great question to study. It would be really interesting to know how the different canonical learning mutants, and also wild types, behave in these experiments before and after a learning session. That will give us a clue on the interaction between learning, individuality and spontaneity. It will be also relevant to study how previous experiences modify personality, and how spontaneity act upon decisions after learning, or if spontaneity is diminished. Kain et al. showed in 2012 that individual flies did not show learning or adaptation during 40 consecutive choice steps in their paradigm, and each light-dark choice was uncorrelated to its previous choice. We are currently building a modified version of Kain et al FlyVac. This apparatus will help us study questions like the one commented by the reviewer. So far, to our knowledge there is no study on this regard. 

- Line 439: I cannot quite certainly follow what was done here; this means I did not get the difference/ relationship between Figures 2C versus D. This would seem essential to judge a key conclusion, though.

Answer: The text was corrected in order to improve the understanding 

- If I understood correctly, it might help to label the X-axis with “Number of times choosing light” or so (I hope I got the sign right here…). Using the abstract expressions “Tube #1, …” would be intuitive only with the suitable graphic/ pictogram.

Answer: The text was corrected in order to improve the understanding, and plots were also modified. 

- Maybe arrange Figure 3 such that the significantly different data are grouped together?

Answer: The plots were rearranged as suggested. 

- Maybe the data from figure 3 would allow some sort of machine-learning approach to see how many “kinds of flies” there are?

Answer: We tried twice to stablish a collaboration in that direction, unfortunately we did not succeed. However, it is for sure one of the following steps. 

- Line 569: It should be “than” and not “then” here.

Answer: The mistake was corrected.

I hope that you will find that this revised and improved version of the manuscript possesses the broad scope, novelty, and high degree of general scientific relevance to warrant publication in PLOS One. I look forward to hearing from you soon.

---

## [Decision Letter · Decision Letter 1]

10 Aug 2021

Collective action or individual choice: Spontaneity and individuality contribute to decision-making in Drosophila

PONE-D-21-09940R1

Dear Dr. Gorostiza,

We’re pleased to inform you that your manuscript has been judged scientifically suitable for publication and will be formally accepted for publication once it meets all outstanding technical requirements.

Kind regards,

Efthimios M. C. Skoulakis, PhD

Academic Editor

PLOS ONE

Additional Editor Comments (optional):

Reviewers' comments:

Reviewer's Responses to Questions

**Comments to the Author**

1. If the authors have adequately addressed your comments raised in a previous round of review and you feel that this manuscript is now acceptable for publication, you may indicate that here to bypass the “Comments to the Author” section, enter your conflict of interest statement in the “Confidential to Editor” section, and submit your "Accept" recommendation.

Reviewer #1: All comments have been addressed

2. Is the manuscript technically sound, and do the data support the conclusions?

Reviewer #1: Yes

3. Has the statistical analysis been performed appropriately and rigorously? 

Reviewer #1: Yes

4. Have the authors made all data underlying the findings in their manuscript fully available?

Reviewer #1: Yes

5. Is the manuscript presented in an intelligible fashion and written in standard English?

Reviewer #1: Yes

6. Review Comments to the Author

Reviewer #1: I am satisfied with the authors edits and additions. Congratulations on your interesting results.

7. PLOS authors have the option to publish the peer review history of their article (what does this mean?). If published, this will include your full peer review and any attached files.

Reviewer #1: No

---

## [Editor Report · Acceptance letter]

17 Aug 2021

PONE-D-21-09940R1 

Collective action or individual choice: Spontaneity and individuality contribute to decision-making in *Drosophila*

Dear Dr. Gorostiza:

I'm pleased to inform you that your manuscript has been deemed suitable for publication in PLOS ONE. Congratulations! Your manuscript is now with our production department. 

Kind regards, 

on behalf of

Dr. Efthimios M. C. Skoulakis 

Academic Editor

PLOS ONE